# What's *Left*? Concept Grounding with Logic-Enhanced Foundation Models

**Joy Hsu***
Stanford University

**Jiayuan Mao***
MIT

**Joshua B. Tenenbaum**
MIT

**Jiajun Wu**
Stanford University

## Abstract

Recent works such as VisProg and ViperGPT have smartly composed foundation models for visual reasoning—using large language models (LLMs) to produce programs that can be executed by pre-trained vision-language models. However, they operate in limited domains, such as 2D images, not fully exploiting the generalization of language: abstract concepts like "*left*" can also be grounded in 3D, temporal, and action data, as in moving to your *left*. This limited generalization stems from these inference-only methods' inability to learn or adapt pre-trained models to a new domain. We propose the **L**ogic-**E**nhanced **F**ounda**T**ion Model (**LEFT**), a unified framework that *learns* to ground and reason with concepts across domains with a differentiable, domain-independent, first-order logic–based program executor. LEFT has an LLM interpreter that outputs a program represented in a general, logic-based reasoning language, which is shared across all domains and tasks. LEFT's executor then executes the program with trainable domain-specific grounding modules. We show that LEFT flexibly learns concepts in four domains: 2D images, 3D scenes, human motions, and robotic manipulation. It exhibits strong reasoning ability in a wide variety of tasks, including those that are complex and not seen during training, and can be easily applied to new domains.

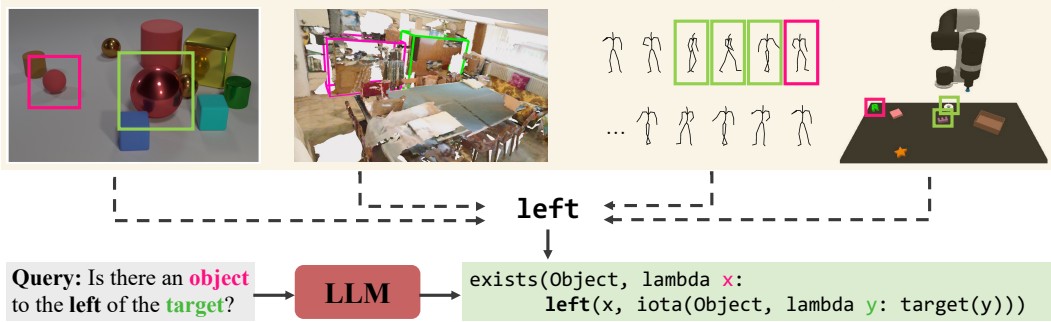

Figure 1: LEFT is a unified concept learning and reasoning framework that grounds modular concepts across domains, and flexibly reasons with concepts across tasks with a foundation model.

## 1 Introduction

The power of language lies in its generalization based on abstraction. A single concept, such as *left*, can be used across domains: in perception, we recognize the *left* leg from a chair's 3D point cloud; in navigation, we go home by turning *left* around the cafe at the end of the block; in manipulation, we pick and place a mug on the *left* of a plate. *Left* has domain-specific groundings in each of these

---

*Equal contribution. Email: `joycj@stanford.edu`

37th Conference on Neural Information Processing Systems (NeurIPS 2023).

domains, but the concept symbol itself as an abstraction serves as a basis for complex and multi-step reasoning across domains.

Machine systems for visual reasoning also benefit from the disentanglement of concept learning and reasoning. Most recently, models such as VisProg [Gupta and Kembhavi, 2023] and ViperGPT [Surís et al., 2023] have leveraged large language models (LLMs) to produce programs based on language queries. For example, to answer "what's the color of the cat?", the program will first locate the cat, then query its color. Such programs are executed with pre-defined and pre-trained vision-language models (e.g., open-vocabulary segmentation models) for grounding and question-answering. These works perform well on image-based reasoning, though surprisingly, despite the generalization seemingly offered for free by language, less success has been shown in other domains.

What's *left* to make visual reasoning systems work across domains? We identify two key shortcomings of existing methods. First, they are *inference-only*, building exclusively upon pre-trained models to interpret concepts such as *left* during program execution. Therefore, in domains where data is scarce (e.g., 3D, human motion, or robot action), pretrained models, if there exist any, do not perform as well. Thus, we need trainable systems that learn the domain-specific groundings of concepts to execute across domains. Unfortunately, these visual reasoning models also cannot be made trainable due to their second shortcoming: they rely on a non-differentiable program executor to compose these grounding modules. Therefore, though *left* may have different groundings in 3D scenes and robot actions compared to 2D images, these modules will not be able to adapt through additional training in those domains.

We address these issues by proposing Logic-Enhanced Foundation Models (LEFT), a model that *learns* to ground and reason with concepts across domains. LEFT also leverages LLMs for language parsing; but unlike prior work, LEFT has trainable concept grounding modules that learn from domain data, made possible via a differentiable, domain-independent, first-order logic–based program executor. Inspired by previous works that combine symbolic programs and deep models via differentiable execution [Mao et al., 2019, Dong et al., 2019], our design no longer requires manually designed domain-specific languages and facilitates generalization across domains. Specifically, LEFT's LLM interpreter takes as input language queries, resolves textual ambiguities, and outputs a non-ambiguous program represented in a general reasoning language of first-order logic, which is shared across domains and tasks. LEFT's executor then executes the logic program with learnable domain-specific grounding modules, which are automatically initialized with concepts generated by the LLM.

LEFT enjoys the advantages of strong performance and data efficiency across domains due to its modular structure. Within a single domain, LEFT generalizes zero-shot to unseen and complex tasks, by leveraging general reasoning capabilities from the LLM interpreter, and effectively recomposing learned, grounded concepts with the generic first-order logic executor. LEFT can be viewed as a generalized framework of VisProg and ViperGPT; in domains where pre-trained models are available and training is not required (e.g., 2D images), LEFT can similarly be used inference-only.

We validate LEFT's performance on four different domains and seven tasks, ranging from 2D question answering, 3D referring expression comprehension, temporal sequence reasoning, and robotic manipulation. A general LEFT model significantly outperforms prior task-specific monolithic methods in performance and in data efficiency settings, and yields comparable performance to prior neuro-symbolic methods while requiring no pre-defined program implementations for each domain. Importantly, the unified LEFT framework can perform concept learning across domains, as well as show zero-shot transfer to three challenging and unseen reasoning tasks. In contrast, inference only LLM-based methods and general vision-language models fail to generalize.

## 2  Related Work

**LLM-based decomposition frameworks.** Our framework integrates neural networks, logic reasoning, and large language models for commonsense reasoning. The first group of related literature studies LLM-based approaches. Inspired by the success of LLMs [Brown et al., 2020], many recent works have proposed frameworks that leverage LLMs to decompose text-based tasks into a sequence of API calls to existing models [Cheng et al., 2023, Dohan et al., 2022, Beurer-Kellner et al., 2023, Zelikman et al., 2023]. Compared to LEFT, these methods run inference only with LLMs and API models, and are limited to the language domain, without learning grounding to modalities. For example, LLMs may be able to reason about categories of objects inferred from language, but it

cannot recognize the candidate objects from the current scenes, or generate robotics movements to move the object. Works such as Gupta and Kembhavi [2023] and Surís et al. [2023] execute programs on images, but assume API access to a set of available modules with no further training. By contrast, LEFT learns to ground modular concepts across different domains by training concept embeddings for each modality, not requiring any pre-defined and pre-trained modules.

**General vision-language models.** In recent years, unified vision-language models (VLMs) have demonstrated success in learning on multimodal domains [Cho et al., 2021, Tsimpoukelli et al., 2021, Wang et al., 2022, Alayrac et al., 2022]. As a representative example, Flamingo [Alayrac et al., 2022] leverages powerful pre-trained vision and language models to achieve few-shot learning on language generation tasks, interleaving images, videos, and text, then outputting associated texts. However, these unified VLMs, and broadly end-to-end methods, are still limited in its application in different domains. In addition, in practice, they struggle with more complex unseen tasks. By contrast, LEFT performs well across domains, and can generalize to new challenging tasks that it has never seen.

**Neuro-symbolic methods.** Neuro-symbolic approaches that combine programs with neural networks have shown success in a variety of visual reasoning domains, with strong data efficiency and generalization abilities from their modular designs. Neuro-symbolic VQA [Yi et al., 2018] proposed symbolic program execution for the question-answering task, and the Neuro-Symbolic Concept Learner [NSCL; Mao et al., 2019] further improved the training paradigm by removing the requirement for dense supervision. Inspired by the success in 2D image domains [Andreas et al., 2016, Johnson et al., 2017b, Han et al., 2019, Hudson and Manning, 2019, Li et al., 2020, Chen et al., 2021a, Mao et al., 2021], neuro-symbolic methods have been introduced for grounding in 3D scenes [Achlioptas et al., 2020, Hong et al., 2022, Hsu et al., 2023], reasoning in temporal sequences [Chen et al., 2021b, Endo et al., 2023], and robotic manipulation [Wang et al., 2023, Mao et al., 2022]. However, these neuro-symbolic works require pre-defined domain-specific languages, such that task instructions are parsed into a constrained set of programs, and each program is then manually implemented in code. Therefore, it is challenging to incorporate commonsense reasoning into them, or generalize to different domains. In contrast, LEFT retains all the benefits of neuro-symbolic learning, while proposing a universal language for all domains. LEFT requires neither any domain-specific definition nor domain-specific program examples. It only leverages structures encoded through minimal prompting examples of first-order logic usage for the LLM; hence, it is generally applicable across domain.

## 3 Logic-Enhanced Foundation Model (LEFT)

Our Logic-Enhanced Foundation Model (LEFT) is a unified framework for concept learning and reasoning across different domains and tasks. It integrates large language models with differentiable logic modules, and modular neural networks for grounding concepts in each modality. Shown in Fig. 2, our system consists of three main components:

1. The first is a domain-independent LLM language interpreter, which generates first-order logic queries to the FOL execution engine (Section 3.1). The generated symbolic programs are represented with a hierarchical first-order logic structure.
2. The second is a domain-independent FOL executor, which executes the logic programs differentiably based on features of entities in the domain: objects, relations, actions, etc. (Section 3.2). LEFT's executor is fully differentiable, which allows for backpropagation.
3. The third is the domain-specific grounding modules, which consist of encoders for each modality that extract entity-centric and relational features, as well as the corresponding concept embeddings implemented as modular neural networks (Section 3.3). The concept grounding modules are initialized with concepts generated by the LLM interpreter.

LEFT is a framework of domain-independent reasoning modules (composed of the LLM interpreter and the FOL executor) and domain-specific grounding modules. With these components, LEFT conducts concept learning and reasoning across domains and tasks.

### 3.1 Domain-independent LLM interpreter

We leverage a pre-trained large language model as our domain-independent language interpreter. It takes language queries as inputs and generates first-order logic queries to downstream modules. The

**Domain-independent**

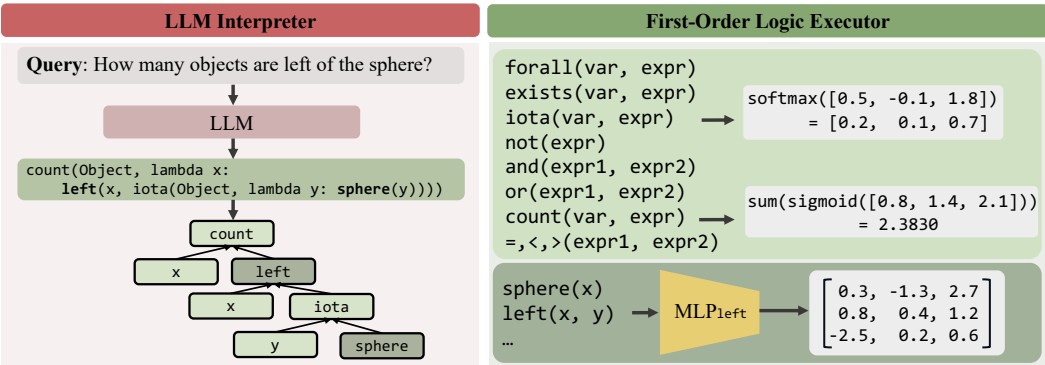

**Domain-specific**

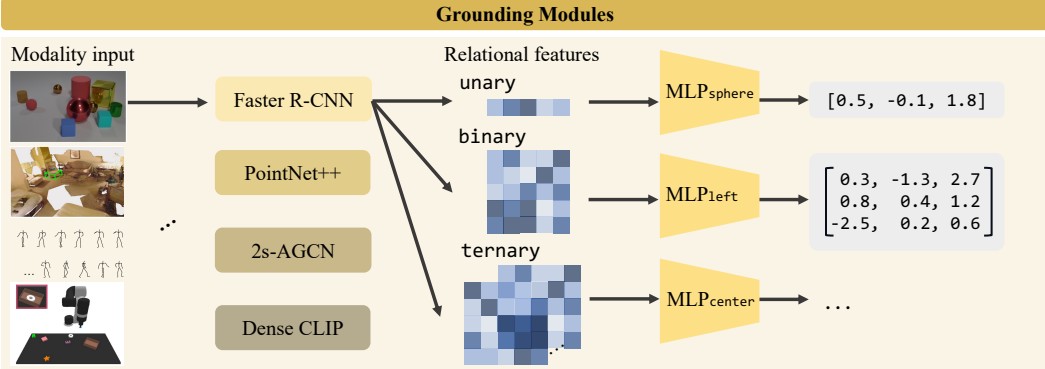

Figure 2: At the core of LEFT is a foundation model-powered language interpreter that generates first-order logic (FOL) programs from language queries, a domain-independent, differentiable FOL executor that operates on all types of entities, and domain-specific grounding modules that connect symbols in the first-order logic language with domain-specific features.

examples we used to prompt the LLMs is a minimal set of general, domain-independent examples and system-level description of syntax rules in our FOL language. The LLM handles both natural language understanding and commonsense reasoning. For example, it needs to resolve coreferences and ambiguous modifier attachments: "Look at the book on the shelf, *the blue one*, select the object left of *it*." Furthermore, it performs commonsense reasoning, such as "Find an object that can fit into a row of blue spheres." The query it generates for the downstream FOL reasoning module will be non-ambiguous programs: $\iota x.blue(x) \wedge sphere(x)$. This strategy maximally harnesses the reasoning capability of LLMs across tasks, while retaining strong generalization. We show in experiments that the LEFT interpreter can reason about analogies, resolve coreferences, and tackle complex puzzles.

Formally, given an input task query $Q$ in language, our LLM interpreter outputs programs $P$, which are programs written in first-order logic that will be passed to downstream FOL executors. Each program is composed of a function name and arguments, and can be chained together hierarchically such that the output of one function can be the input argument to another function. The LLM-interpreted program contains domain-independent FOL functions, which can take outputs from domain-specific functions as arguments. The FOL functions are a general set of pre-defined programs with built-in implementations and are listed in full in Table 1. These functions are either logic and numeric operations (such as counting, forall, exists), or functions that handle inputs and outputs (e.g., return a text description of object categories, or execute an action output by other modules). We present the syntax and built-in components of our FOL language in the next section.

LEFT leverages GPT-3.5 [Brown et al., 2020] as its LLM backbone. Importantly, the LEFT prompts are domain-independent, including simple syntax rules and examples written with general concepts such as *cat* and *apple* that do not occur in the any of our evaluation domains. Examples in our prompt also have minimal complexity, such as first-order logic sentences for "*Is there an apple next to the cake*", paired with short descriptions of FOL usage, such as "*to classify whether two objects have a certain property, use 'on(x, y)'.*" Additionally, we use a step-by-step prompting paradigm, which

| Name | Syntax | Description |
|------|--------|-------------|
| variables | $x, y, z, \cdots$ | Variables that refer to entities in the domain. |
| $\exists, \forall$ | $exists(var, expr)$ | *expr* is an expression that contains the variable *var*; return true if there is at least one assignment of *var* that satisfies *expr*. |
| iota ($\iota$) | $iota(var, expr)$ | *expr* is an expression that contains the variable *var*; return an assignment to *var*'s that satisfies *expr*. |
| not | *not expr* | Compute the negation of an expression. |
| and, or | $expr_1$ *and* $expr_2$ | Compute the conjunction/disjunction of an expression. |
| count | $count(var, expr)$ | Count the number of assignments to *var* that will make *expr* true. |
| $=, <, >$ | $eq(expr_1, expr_2)$ | Built-in number comparison functions. |
| view | $view(expr)$ | *expr* is an object, e.g., $view(\iota x.blue(x))$; it changes the view direction to face the object. |
| describe | $describe(expr)$ | *expr* is a text description, e.g., $describe(\iota c.color(c, \iota x.sphere(x)))$; it returns the description (the color of the sphere). |
| do | $do(expr)$ | *expr* is an action, e.g., $do(\iota a.pick\text{-}up(a, \iota x.sphere(x)))$; it executes the action (pick up the sphere). |
| concepts | $left(x, y)$ | A function that takes objects, actions, and texts as inputs and returns Boolean values. |

Table 1: LEFT's first-order logic language, a general reasoning language across domains.

asks the LLM to first reason about the query and simplify the text in language form, then generate FOL programs.

## 3.2 Domain-independent first-order logic executor

The LEFT domain-independent executor operates on first-order logic, which serves as a general reasoning language shared across different tasks. Given the parsed FOL programs $P$ from the LLM interpreter, the executor operates $P$ with grounding modules to return the final answer. The programs are executed recursively to model the hierarchical structure of the FOL reasoning process. The LEFT programs are implemented as differentiable first-order logic modules in Python code.

Our FOL language contains three parts: first, basic first-order logic operations such as Boolean operations and logic quantifiers; second, built-in functions such as *describe*, which generate a natural language response for an entity or a pair of entities, and *do*, which execute a robotic action; and third, domain-specific concept names, including object properties (e.g., *cat*), object relations (e.g., *left*), and actions (e.g., *pack*). These concept predicates are automatically produced by the LLM based on the language query, instead of being selected from a pre-defined set of functions; hence, the expressiveness of LEFT is not limited by the given set of primitives such as exists and count. It can learn to execute any new functions proposed by the LLM.

For a given universe of entities, the LEFT executor executes all expressions using a tensor-based representation of truth values. For example, given a scene with $N$ objects, the execution result of the expression $sphere(x)$, where $x$ is an unquantified variable, can be represented as a vector of length $N$, where each entry $i$ denote whether object $i$ is a sphere. Relation expressions such as $left(x, y)$ can be represented as matrices of size $N \times N$. Whether an object has been selected can be represented similarly, as a vector of size $N$, where each entry $i$ is the probability of object $i$ being selected. In practice, values such as $sphere(x)$ and $left(x, y)$ will be predicted by domain-specific grounding neural networks. In Table 1, we detail the execution strategy for all operations. Essential to our goal of a general neuro-symbolic framework, LEFT only requires these listed FOL functions to reason across numerous domains, including 2D, 3D, temporal, and robotic manipulation.

The programs generated from our LLM interpreter include FOL programs along with domain-specific concepts, which are executed by our grounding modules. The LEFT executor implements these programs based on the arity (e.g., number of arguments in a function) of the first-order logic program. Unary-level functions are implemented with entity-centric features of size $N$ for each entity from the input, and binary-level functions are implemented with relational features of size $N \times N$, etc. Notably, each domain-specific concept is initialized by the LLM, and the function implementation is executed with modular neural networks, which does not require manual definition in code.

**Execution.** We describe implementations for the FOL programs below. See Appendix for details.

*exists(var, expr)*: We first recursively execute *expr* and get a tensor where *var* is one of the dimensions of the tensor. We perform a max pooling over that dimension. For example, the execution of $sphere(x)$ yields a vector of length $N$, for $exists(x, sphere(x))$, we take the max value of the returned tensor.

*iota(var, expr)*: We recursively execute *expr* and get a vector where *var* corresponds to the dimensions of the vector. We then perform a softmax operation over the vector.

*and(expr$_1$, expr$_2$)*: The recursive execution of $expr_1$ and $expr_2$ will yield two tensors. The conjunction of two resulting tensors is done by taking the element-wise minimum operation for two tensors. The same execution strategy is used for or operations (by taking the element-wise maximum) and logical negation (by taking the element-wise negation).

*count(var, expr)*: We recursively execute *expr*, then perform a sigmoid operation followed by a sum over the vector to compute the expected number of objects.

*eq(expr$_1$, expr$_2$)*: The recursive execution of $expr_1$ and $expr_2$ will yield two scalar values $s_1$ and $s_2$. We define the probability of $s_1 = s_2$ as $\sigma\left(\alpha \cdot (\gamma - |s_1 - s_2|)\right)$, where $\alpha = 8$ and $\gamma = 0.25$ are fixed hyperparameters. Less than and greater than operations are implemented similarly.

*left$(x, y)$*: As an example of a domain-specific concept, the *left* function takes two variables as its input, and generates a score matrix of size $N \times N$. This is done by applying *MLP$_{left}$* on the binary embedding extracted by the feature extractor. Note that throughout the execution, we keep the "logit" of the predicted values instead of normalizing them to $[0, 1]$ with a sigmoidal activation. This improves the numerical stability of the execution.

## 3.3 Domain-specific grounding modules

The final component of LEFT is the domain-specific grounding modules that connect concepts in the first-order logic language with modality-specific representations. LEFT does not depend on a pre-defined set of lexical concepts, and relies on LLMs to automatically extract concept names from language in the data. Our framework then initializes domain-specific grounding modules accordingly to learn the grounding of these concepts from training data. In other words, the grounding modules do not require manual definitions, can support execution of any given query, and are realized with a generic mechanism described below. The grounding modules are implemented with modular neural networks, consisting of modality-specific encoders and concept embeddings. We first extract entity-centric and relational features with modality-specific encoder $\varepsilon$. For 2D images, $\varepsilon$ is Faster R-CNN [Ren et al., 2015] for encoding object-centric features and DenseCLIP [Rao et al., 2022] for generating object attention masks. For 3D point clouds, $\varepsilon$ is PointNet++ [Qi et al., 2017]; and for motion sequences, $\varepsilon$ is the Two-Stream Adaptive Graph Convolutional Network [Shi et al., 2019]. Unary features are represented as a set of features for each entity, binary features as a 2D matrix of relations between each pair of entities, and ternary relations as a 3D matrix, etc.

Given entity-centric representations, we then execute concept functions such as *sphere* and *left*(x, y). In general, we can have arbitrary neural networks that take the relational representation of entities and outputs a (soft) Boolean value from $[0, 1]$. In this paper, we initialize a multi-layer perceptron (MLP) for each concept that has been mentioned by the large language model, to represent the concept embedding. For example, in our 2D image domain, each object $i$ has a vector embedding $obj_i$ extracted by a Faster R-CNN encoder. We apply *MLP$_{sphere}$* on each object representation $obj_i$ and generate a vector of length $N$, which is the execution result for the expression $sphere(x)$.

The grounding modules are automatically initialized based on the concept name as well as *arity* proposed by the LLM. For example, if a concept is unary (e.g., *red*(x)), the grounding module will be a MLP layer that maps object representations to the aforementioned soft Boolean value scalar. For the concept *beside*, the LLM specifies that the concept takes in features of two entities, and hence LEFT creates an MLP that operates on binary features. The domain-specific feature extractors and concept MLPs are trained through backpropagation, as our executor is fully differentiable.

## 4 Experiments

We evaluate LEFT on four domains and seven tasks, and show its effectiveness in multiple settings. In Section 4.1, we compare LEFT to prior state-of-the-art neuro-symbolic and monolithic end-to-end

| **2D question answering** | **3D referring expression** | **Temporal sequence reasoning** | **Robotic manipulation** |
|---|---|---|---|
| 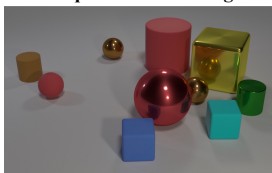 | 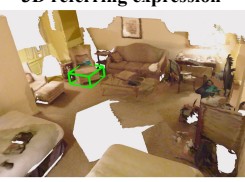 | 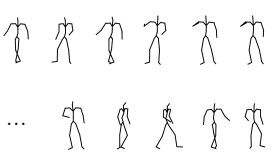 | 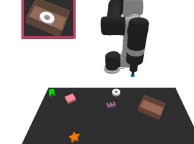 |
| **Query**: Are there two objects of the same color nearly touching? | **Query**: Choose the ottoman sitting next to the end table. | **Query**: What body part does the person use before they turn? | **Query**: Pack the ring into the brown box. |

Figure 3: LEFT is evaluated on four broad categories of tasks, ranging from concept learning in 2D images, 3D point clouds, temporal motion sequences, and robotic manipulation.

methods on concept learning across four domains. Our model excels at task performance and also data efficiency, without requiring any domain-specific knowledge. Next, in Section 4.2, we present LEFT's zero-shot generalization performance to three novel and challenging visual reasoning tasks, and compare its reasoning capabilities to that of LLM-based decomposition frameworks and a general vision-language model. Implementation and training details for LEFT can be found in the Appendix.

## 4.1 Concept learning across domains

We train and validate LEFT on six different datasets, grouped into four domains (Figure 3).

**2D visual concept learning.** In the first domain, we focus on the task of visual question answering in 2D images: given an image and a language query (e.g., *"How many small cylinders are partially hidden?"*), the goal is to predict an answer from a set vocabulary. The accuracy is calculated as the exact match to ground truth answers. We validate LEFT on the CLEVR and CLEVR-Humans dataset [Johnson et al., 2017a], which contain images with objects rendered of different shape, size, material, and color. CLEVR contains synthetically generated queries, while the CLEVR-Humans is a human-annotated dataset with diverse natural language questions.

**3D visual concept learning.** The second domain evaluates 3D referring expression comprehension. The input is a set of object point clouds in the scene, and a query (e.g.*"The monitor that you would class as in the middle of the other two"*). Models are evaluated by the accuracy of selecting the correct target object. In the 3D domain where we need reference frames to disambiguate relations like *left*, the grounding of such concepts will consider the viewpoint specified in the language query, through a generic high-arity function that does not require a specific implementation. We evaluate LEFT on the ReferIt3D dataset [Achlioptas et al., 2020], with 3D point clouds from ScanNet [Dai et al., 2017], a collection of indoor scenes. ReferIt3D consists of two subsets, SR3D, which focuses on spatial reference with synthetically generated queries, and NR3D, with human-annotated queries. We use the NR3D subset from NS3D, which contain the same set of concepts as in SR3D.

**Motion capture concept learning.** The third domain contains human motion capture. In particular, we focus on the task of question answering in temporal motion sequences. Given a sequence of human motion, parameterized by joints, and a question (e.g.*"What action does the person do before they move backward and after they kick?"*), the goal is to predict the exact match answer. We train and evaluate LEFT on the HumanMotionQA dataset [Endo et al., 2023], which asks questions about temporal relations and a variety of attributes, including action, direction, and body parts.

**Robotic manipulation.** Finally, we demonstrate that LEFT can learn action concepts that correspond to robotic motions, in the context of robotic manipulation. The input to this task is an image representing a tabletop environment, and a language instruction (e.g., *"Pack the yoshi figure into the brown box."*) The goal is to generate a sequence of actions that maximizes the average total reward over this robotic manipulation task. We validate LEFT on the CLIPort dataset [Shridhar et al., 2022], focusing on the task of sequential packing Google Scanned Objects [Downs et al., 2022].

### 4.1.1 Accuracy

We compare LEFT against two categories of top-performing models across all six datasets. The first category contains neuro-symbolic models, which require a pre-defined domain-specific language and implementations for each program. The second category contains monolithic end-to-end methods which usually require a significant amount of data to train. Notably, our unified LEFT framework is able perform on all domains, instead of requiring specific reasoning modules for each setting.

|  | Pre-def. | CLEVR | SR3D | HumanMotionQA | Cliport |
|---|---|---|---|---|---|
| Prior NS | Yes | 0.992 (NSCL) | 0.627 (NS3D) | 0.578 (NSPose) | 0.797 (ProgramPort) |
| End-to-End | No | 0.989 (MAC) | **0.670** (BUTD [†]) | 0.430 (MotionCLIP) | 0.708 (Cliport) |
| LEFT | No | **0.996** | 0.620 | **0.582** | **0.793** |

Table 2: Results comparing LEFT to prior top-performing neuro-symbolic and end-to-end works. Our unified LEFT framework yields strong performance on all settings without any pre-defined programs.

Table 2 summarizes the results of different models on four datasets that involve synthetically generated language, and hence ground truth programs exist for neuro-symbolic methods and our approach. We see that across CLEVR, SR3D, HumanMotionQA, and CLIPort, LEFT achieves comparable performance to prior neuro-symbolic works [Mao et al., 2019, Hsu et al., 2023, Endo et al., 2023, Wang et al., 2023], and outperforms end-to-end methods in CLEVR, HumanMotionQA, and CLI-Port [Hudson and Manning, 2018, Jain et al., 2022, Shridhar et al., 2022]. Importantly, our LEFT model applies a unified language interpretation and program execution framework for across all domains and tasks, while prior work on neuro-symbolic learning requires domain-specific languages and reasoning modules for each dataset. This suggests that first-order logic language can serve as an effective and general language for representing language queries.

In Table 3, we examine LEFT on tasks with human-annotated natural language queries. The first number for LEFT represents accuracy out of all queries where the program generated by LLM is executable, while the latter number represents accuracy out of all queries, considering the percentage of LLM outputs that are not executable.

|  | Pre-def. | CLEVR-Humans | NR3D |
|---|---|---|---|
| Prior NS | Yes | 0.678 (NSVQA) | 0.526 (NS3D) |
| End-to-End | No | **0.817** (MDETR) | 0.430 (MVT) |
| LEFT | No | 0.788 / 0.698 | **0.496 / 0.485** |

Table 3: Comparisons on natural language tasks.

We note that MDETR [Kamath et al., 2021] outperforms LEFT on CLEVR-Humans, possibly because it can better capture the similarity between similar concepts such as *front* and *in-front-of*. In contrast, the representations for different concepts in LEFT are independent. The prior neuro-symbolic method we compared against outperforms LEFT on NR3D, as it also leverages an LLM for language interpretation, but uses a set of domain-specific examples in prompts that are hand-crafted.

### 4.1.2 Data efficiency

We evaluate LEFT on data efficiency settings for CLEVR [Johnson et al., 2017a] and SR3D [Achlioptas et al., 2020]. LEFT significantly outperforms all end-to-end methods at every data efficient percentage point tested, and performs comparably to best-performing neuro-symbolic methods [Hsu et al., 2023, Jain et al., 2022, Huang et al., 2022, Yang et al., 2021, Mao et al., 2019, Mascharka et al., 2018, Hudson and Manning, 2018] (See Table 4 and Table 5). The percentages indicate the percentage of train data used for training compared to the full dataset.

|  | Pre-def. | 0.5% | 1.5% | 2.5% | 5% | 10% | 100% |
|---|---|---|---|---|---|---|---|
| NS3D | Yes | 0.426 | 0.520 | 0.556 | 0.591 | 0.600 | 0.627 |
| BUTD-DETR | No | 0.083 | 0.158 | 0.259 | 0.395 | 0.528 | **0.670** |
| MVT | No | 0.161 | 0.275 | 0.322 | 0.380 | 0.491 | 0.645 |
| SAT | No | 0.172 | 0.260 | 0.298 | 0.330 | 0.362 | 0.579 |
| LEFT | No | **0.410** | **0.521** | **0.565** | **0.579** | **0.591** | 0.620 |

Table 4: Results on SR3D in data efficient settings. LEFT performs comparably to prior neuro-symbolic works, and outperforms end-to-end works significantly.

Focusing on CLEVR as an ablation study, we note that NSCL outperforms LEFT on the 10% setting. We attribute this gap to the additional domain-specific inductive biases that NSCL leverages. In particular, NSCL assumes the existence of four attribute spaces (color, shape, material, and size), and each concept lies in exactly one attribute space. Furthermore, all concepts within the same attribute

---

[†]BUTD operates on a constrained setting, assuming 485 classes instead of the full 607 classes in ReferIt3D.

space are mutually exclusive. Therefore, the inductive biases encoded by NSCL can be seen as a combination of a hierarchical program-based reasoning mechanism, and additional domain-specific priors on concepts. By contrast, LEFT only leverages the program-based reasoning mechanism, which is general across many domains and tasks. In comparison of our model with other baselines and NSCL, we show that this general program-based reasoning mechanism is powerful enough to explain most of the data efficiency improvements.

## 4.2 Reasoning generalization across tasks

Finally, we present LEFT's ability to zero-shot generalize to unseen tasks.

**CLEVR reasoning tasks.** We generate three zero-shot transfer tasks based on the CLEVR datasets [Johnson et al., 2017a], with 100 evaluation examples each. See Figure 4 for an example of each task. The CLEVR-

|         | Pre-def. | 10%   | 25%   | 50%   | 100%  |
|---------|----------|-------|-------|-------|-------|
| NSCL    | Yes      | 0.989 | 0.992 | 0.994 | 0.992 |
| TbD-Net | Yes      | 0.541 | 0.560 | 0.912 | 0.991 |
| MAC     | No       | 0.673 | 0.940 | 0.974 | 0.989 |
| LEFT    | No       | **0.941** | **0.991** | **0.992** | **0.996** |

Table 5: Results on CLEVR in data efficient settings.

Ref task consists of a CLEVR image and language queries, requiring LEFT to locate the exact target object. The CLEVR-Puzzle task consists of images paired with visual puzzle descriptions, which necessitates a variable assignment puzzle solving. The CLEVR-RPM task is based on Raven's Progressive Matrices, which consists of an image and a 3x3 grid describe by language. This task requires the LLM interpreter to reason about patterns in the grid by row and column to determine the attributes of the target object, and generate queries to the FOL executor to check if such an object exists in the scene. We include dataset details in the Appendix.

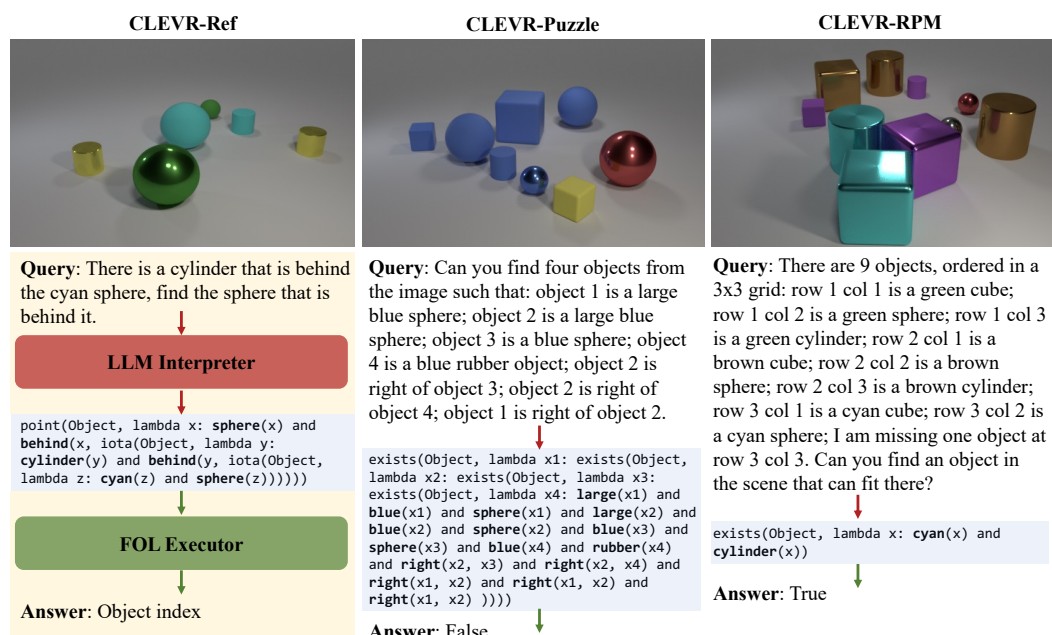

Figure 4: The three zero-shot generalization tasks. As seen in the CLEVR-Ref example, LEFT's domain-independent modules resolve the language into first-order logic and generate the answer.

### 4.2.1 Accuracy

We train LEFT on the CLEVR question answering task, and report zero-shot transfer performance on the three unseen reasoning tasks. In Table 6, we see that LEFT achieves strong performance on all datasets. We compare LEFT performance with ground truth programs (GT programs) and with FOL programs from our LLM interpreter (LLM programs). This ablation demonstrates that there are only small drops in performance with LLM programs as compared to ground truth programs across all three complex reasoning tasks. LEFT can zero-shot transfer to novel visual reasoning tasks with LLM generated first-order logic, and effectively reuse concept embeddings, enabling flexible

|                                                | CLEVR-Ref | CLEVR-Puzzles | CLEVR-RPM |
|------------------------------------------------|-----------|---------------|-----------|
| LEFT + GT programs                             | 1.00      | 0.92          | 1.00      |
| LEFT + LLM programs                            | **0.94**  | **0.75**      | **0.87**  |
| VisProg [Gupta and Kembhavi, 2023]             | 0.35      | 0.27          | 0.51      |
| ViperGPT [Surís et al., 2023]                  | 0.08      | 0.34          | 0.04      |
| OpenFlamingo 4-shot [Awadalla et al., 2023]    | N/A       | 0.54          | 0.52      |
| OpenFlamingo 8-shot                            | N/A       | 0.57          | 0.52      |

Table 6: Zero-shot generalization of LEFT to three unseen tasks; LEFT recomposes domain-independent programs and learned concepts with the LLM interpreter for flexible transfer.

generalization. Importantly, we see that LEFT performs well on CLEVR-RPM, a challenging task which contains complex Raven's Progressive Matrices [Barrett et al., 2018] descriptions that require the LLM interpreter to reason about patterns described by language. While the prompts given to the LLM are simple, our interpreter can solve complex tasks.

As baselines, we compare LEFT with LLM-based decomposition frameworks and general vision-language models. VisProg [Gupta and Kembhavi, 2023] and ViperGPT [Surís et al., 2023] are inference only, integrated LLM with APIs frameworks, that has shown success on a variety of visual reasoning tasks. In Table 6, we see that LEFT significantly outperforms VisProg and ViperGPT on the CLEVR transfer tasks. The LLM-based decomposition frameworks often fail to output executable code for complex queries,; in addition, the accuracy of execution by the APIs is low. For example, on the CLEVR-RPM task, only $0.40$ of queries were executed successfully by ViperGPT, with accuracy of $0.10$ out of executable queries. As these methods execute programs with a constrained set of pre-defined functions, the accuracy for executing undefined relations, such as *to the right of*, which is learned in LEFT, is low. In addition, for complex domains such as 3D and temporal human motion sequences, it is unclear what pre-defined models should be used for these inference-only frameworks.

For comparison to general vision-language models, we evaluate OpenFlamingo [Awadalla et al., 2023], an open-sourced version of Flamingo [Alayrac et al., 2022]. OpenFlamingo is a VL model backed by a CLIP vision encoder [Radford et al., 2021] and a LLAMA language encoder [Touvron et al., 2023]. On both 4- and 8-shot variants, LEFT outperforms OpenFlamingo significantly. Notably, OpenFlamingo cannot predict answers for CLEVR-Ref, which requires an object index as output.

## 5 Discussion

We have presented the Logic-Enhanced Foundation Model (LEFT), a unified concept learning and reasoning framework that flexibly learns concepts across different domains, and generalizes reasoning to unseen, complex tasks. Our framework combines foundation models and first-order logic executors as domain-independent reasoning modules, and learnable grounding modules for specific domains. LEFT represents a step towards modeling general reasoning capability in a rightfully grounded way.

LEFT leverages pre-trained LLMs for domain-independent reasoning. Hence, the results of our system are limited by the quality of the LLM, which we cannot finetune. For syntax errors in LLM-generated programs, we can easily detect them and resample. However, we currently cannot recover from semantic errors in the programs. A future direction would be to incorporate execution traces or failures as additional inputs to LLMs such that they can fix the queries they generate.

LEFT may also fail with insufficient domain data, if it cannot effectively train the domain-specific grounding modules. We note that this is a limitation for most learning systems trained solely on one given dataset. To partially mitigate this issue, LEFT instructs LLMs to canonicalize concepts when reasonable, which helps learning certain rare concepts. It is also straightforward to incorporate externally trained modules, by directly integrating pretrained visual models as domain-specific grounding modules. For example, we can plug in Grounding DINO for object detection if we have prior task knowledge [Liu et al., 2023]. However, for some domains, there are no off-the-shelf visual recognition models (e.g., relations and human motions); therefore, LEFT can train its concept groundings by learning from data. Notably, LEFT is also more data efficient compared to end-to-end methods due to its modular structure, as it can decompose complex concepts into primitive ones.

Our framework currently does not model the interaction between the language and perception. Instead, LEFT relies on context-dependent grounding modules which learn from data to resolve ambiguous language. Future directions for LEFT include building in pragmatic reasoning frameworks, modelling speaker intents, and leveraging feedback from perception execution.

**Acknowledgments.** We thank Weiyu Liu and Gabriel Poesia for providing valuable feedback on the paper. This work is in part supported by the Stanford Institute for Human-Centered Artificial Intelligence (HAI), ONR MURI N00014-22-1-2740, ONR N00014-23-1-2355, Air Force Office of Scientific Research (AFOSR) YIP FA9550-23-1-0127, the Center for Brain, Minds, and Machines (CBMM), the MIT Quest for Intelligence, MIT–IBM Watson AI Lab, Analog Devices, Autodesk, J.P. Morgan, and Salesforce. JH is supported by the Knight Hennessy Scholarship and the NSF Graduate Research Fellowship.

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

# Supplementary for What's *Left*? Concept Grounding with Logic-Enhanced Foundation Models

The appendix is organized as the following. In Appendix A, we formally define the first-order logic programs used by LEFT as well as discuss the broader impact of our method. In Appendix B, we present experiment results with error bars and details on the construction of zero-shot transfer tasks. In Appendix C, we provide implementation, training details, and code.

## A    LEFT

### A.1    Function definitions

We summarize the function definitions of LEFT in Table 7. We note that $\mathcal{E}(expr)$ represents the execution result of *expr*. If the input to *concept* functions contains a concrete entity (e.g., result of *iota*), we take the dot product between the entity and the MLP result. If the argument is text (e.g., color), the function is implemented as a generative function instead of a discriminative one. Ternary and high-arity relations can be generalized similarly to unary and binary relations.

| Signature & Implementation | Semantics |
|---|---|
| *exists(var, expr)* $\longrightarrow$ *b: boolean* 
 **Implementation:** $b = max(\mathcal{E}(expr))$ | Return true if there is at least one assignment of *var* that satisfies *expr*. |
| *forall(var, expr)* $\longrightarrow$ *b: boolean* 
 **Implementation:** $b = min(\mathcal{E}(expr))$ | Return true if there is all assignments to *var* satisfy *expr*. |
| *iota(var, expr)* $\longrightarrow$ *e: entity* 
 **Implementation:** $e = softmax(\mathcal{E}(expr))$ | Return an assignment to *var*'s that satisfies *expr*. |
| *not(expr)* $\longrightarrow$ *e: boolean* 
 **Implementation:** $e_i = -\mathcal{E}(expr)_i$, for all $i \in \{1, 2, \cdots, N\}$ | Compute the negation of an expression. |
| *and(expr$_1$, expr$_2$)* $\longrightarrow$ *e: boolean* 
 **Implementation:** $e_i = min(\mathcal{E}(expr_1)_i, \mathcal{E}(expr_2)_i)$, for all $i \in \{1, 2, \cdots, N\}$ | Compute the conjunction of two expressions. |
| *or(expr$_1$, expr$_2$)* $\longrightarrow$ *e: boolean* 
 **Implementation:** $e_i = max(\mathcal{E}(expr_1)_i, \mathcal{E}(expr_2)_i)$, for all $i \in \{1, 2, \cdots, N\}$ | Compute the disjunction of two expressions. |
| *count(var, expr)* $\longrightarrow$ *c: integer* 
 **Implementation:** $c = \sum_i [\sigma(\mathcal{E}(expr)_i)]$ | Return the count of assignments to *var*'s that satisfies *expr*. |
| *greater_than(expr$_1$, expr$_2$)* $\longrightarrow$ *b: boolean* 
 **Implementation:** $b = \sigma(\tau \cdot (\mathcal{E}(expr_1) - \mathcal{E}(expr_2) - \gamma))$ | Return a score indicating whether $\mathcal{E}(expr_1)$ is greater than $\mathcal{E}(expr_2)$. |
| *view(expr)* 
 **Implementation:** $\mathcal{E}(expr)$ | Set the object returned by $\mathcal{E}(expr)$ as the object that the agent is looking at. |
| *do(expr)* 
 **Implementation:** $\mathcal{E}(expr)$ | Perform action computed by $\mathcal{E}(expr)$. |
| *describe(var, expr)* $\longrightarrow$ *e: text* 
 **Implementation:** $e = softmax(\mathcal{E}(expr))$ | Return the best assignment to *var* (e.g., colors, shapes, etc.) that describes the object returned by $\mathcal{E}(expr)$. |
| *concept(var$_x$)* $\longrightarrow$ *e: boolean* 
 **Implementation:** $e_i = \text{MLP}^{\text{concept}}\left(f_i^{\text{unary}}\right)$ | A unary function that takes objects, actions, and texts as inputs and returns Boolean values. |
| *concept(var$_x$, var$_y$)* $\longrightarrow$ *e: boolean* 
 **Implementation:** $e_{i,j} = \text{MLP}^{\text{concept}}(f_{i,j}^{\text{binary}})$ | A binary function that takes objects, actions, and texts as inputs and returns Boolean values. |

Table 7: Function definitions for first-order logic programs in LEFT.

## A.2 Broader impact

LEFT leverages a pre-trained large language model as its language interpreter, and hence, even though our prompts are general examples of first-order logic, we do not have direct control over the LLM's generation. The LLM may output harmful biases, which we will highlight as an important warning in our code release.

# B Experiments

## B.1 Error bars

We present LEFT performance with error bars, taken from three runs with different seeds. The results for CLEVR [Johnson et al., 2017a], CLEVR-10%, and CLEVR-Humans are below. All values are averaged over three random seeds, and the $\pm$ sign shows the computed standard error.

|      | CLEVR-10% | CLEVR-100% | CLEVR-Humans |
|------|-----------|------------|--------------|
| LEFT | $0.942_{\pm 0.008}$ | $0.996_{\pm 0.001}$ | $0.780_{\pm 0.003}$ / $0.694_{\pm 0.001}$ |

Table 8: Results on the CLEVR dataset with standard errors. Average performance and standard errors are computed based on three random seeds.

## B.2 Transfer task construction

We detail the construction processes for CLEVR-Ref, CLEVR-Puzzles, and CLEVR-RPM below.

**CLEVR-Ref**  We generate the CLEVR-Ref dataset using a subset of the templates from the CLEVR-Ref+ [Liu et al., 2019] dataset. We did not directly use the CLEVR-Ref+ dataset because it contains ordinal number concepts (in particular, the first, the second, etc.) that does appear in the original CLEVR dataset. Concretely, we used the following four templates:

1. {Select, Find, Point to} the *Object 1*.
2. {Select, Find, Point to} the *Object 1* (that is) *Relation 1* the *Object 2*.
3. {Select, Find, Point to} the *Object 1* (that is) *Relation 1* the *Object 2* and *Relation 2* the *Object 3*.
4. There is a(n) *Object 2* (that is) *Relation 2* the *Object 3*, {select, find, point to} the *Object 1* (that is) *Relation 1* it.

Here, items such as *Object 1*, *Object 2*, and *Object 3* will be replaced by object-level descriptors such as "shiny cube." *Relation 1* and *Relation 2* will be replaced by relational concepts in the dataset.

**CLEVR-Puzzles**  The CLEVR-Puzzles dataset is generated based on the following template: "Can you find four objects from the image such that: *Object i* is *description*; *Object i* is *relation object j*." Here, *Object i*, *object j* will be replaced by "Object 1," "Object 2," "Object 3," or "Object 4;" descriptions will be replaced by object-level descriptions such as "shiny cube;" relations will be replaced by relational concepts in the dataset. In our data generation, we use four object descriptions (one for each object) and three object relation descriptions. Furthermore, we make sure that none of the object-level descriptions can uniquely identify an object, but the global solution is unique.

**CLEVR-RPM**  The CLEVR-RPM dataset is generated based on the following template: "There are 9 objects, ordered in a 3x3 grid: row 1 col 1 is a *description 1*; row 1 col 2 is a *description 2*; row 1 col 3 is a *description 3*; row 2 col 1 is a *description 4*; row 2 col 2 is a *description 5*; row 2 col 3 is a *description 6*; row 3 col 1 is a *description 7*; row 3 col 2 is a *description 8*; I am missing one object at row 3 col 3. Can you find an object in the scene that can fit there?" Here, descriptions are object-level descriptions such as "shiny cube." Following existing work on Raven's Progressive Matrices [Barrett et al., 2018], we randomly choose the "progression dimensions" from color, shape, material, and size for rows and columns.

# C   Model

## C.1   Implementation details

LEFT consists of several modular components in code. The first is a GPT-3.5 [Brown et al., 2020] backed language interpreter, which takes as input queries from each dataset, and outputs first-order logic. The domain initializer takes the generated first-order logic programs and automatically initializes concept embeddings based on the parsed arity. The domain-specific grounding modules extract entity and relational features from the input modality. Finally, the domain-independent executor executes differentiable logic programs with the outputs of the grounding modules. We release our code below.

## C.2   Training details

We use the official data splits released for each dataset, CLEVR [Johnson et al., 2017a], ReferIt3D [Achlioptas et al., 2020], HumanMotionQA [Endo et al., 2023], and Cliport [Shridhar et al., 2022]. The core hyperparameters were set as 128 for concept embedding dimensions, and learning rate taken from prior neuro-symbolic concept learning repositories that we use as baselines. The full details are provided with the code, linked below.

## C.3   Compute

We trained with 1 NVIDIA Titan RTX per experiment for all datasets, from an internal cluster.

## C.4   Code

We publicly release our code. See our project website for additional details.

