# OpenReview forum: "What’s Left? Concept Grounding with Logic-Enhanced Foundation Models"
_NeurIPS.cc/2023/Conference — NeurIPS 2023 poster_

### Official Review · Reviewer_DugZ · 2023-06-28

**Soundness:** 2 fair
**Presentation:** 3 good
**Contribution:** 4 excellent
**Rating:** 6
**Confidence:** 3

**Summary:**

This paper introduces Neuro-FOL, a method combining a large language model for generating FOL programs, an FOL executor and trainable concept grounding modules to improve performance on a number of visual/3D QA style tasks.

In the method, a LLM interpretor is used to generate a FOL program that is not specific or grounded to the exact domain. A differentiable first-order logic executor is then used to hierarchically evaluate the value of the program by using learnable, domain-specific grounding modules to ground the program in the specific domain.

The paper evaluates in 4 settings: 2D question answering, 3D referring expressions, temporal sequence reasoning and robotic manipulation. The method performs similarly to both NS and end-to-end methods in the normal data settings shown in Table 2, beats end-to-end methods in low data settings (Table 4). The paper also shows good performance on a set of held-out CLEVR tasks.

**Strengths:**

I think main idea of the paper is really clever and well motivated. I think using a pre-trained LLM to do the logical breakdown of the query by generating a program, which can then be differentially trained is really smart. It gives you a way to have both the generality you get from LLMs having seen many concepts and the flexibility of adapting to a new domain.

The wide variety of domains and datasets evaluated is really good. Shows that the method works across a wide variety of different domains.

Shows really good performance compared to end-to-end methods in the rare data case, which is good.

Paper is pretty clearly presented and motivated.

**Weaknesses:**

I think the claim of it being a universal language for all domains and therefore generally applicable across domains is maybe misleading and not totally supported. I think I agree that you could generate a program for pretty much any query-based problem, but it sort of missing a critical assumption, that the method has sufficient data to train each of the domain-specific module that you would generate. And of course, if during evaluation, if the program generated a module that it hadn't seen during training, it would fail. I think this is a real limitation of the method that should be more clearly mentioned in the paper.

Related to the above point, the set of domains is somewhat limited, and especially the kinds of reasoning here are pretty narrow. For instance, the 2D question answering is all done on CLEVR where the visually domain is extremely limited (only a set number of different block objects in a set number of colors) and a rather limited space of kinds of questions (questions chaining together reasoning about objects color, relative position, count etc). There are for instance many kinds of questions about images (for instance in the VQA dataset) that might not really have good coverage in the training set to work. If you had to train a new MLP classifier for each kind of attribute, object class and relation you might find in VQA, it's not totally obvious to me that this method would have enough data to actually work.  SR3D is definitely more visually diverse in the number of object types, but the question types are also quite limited to these kinds of spatial reasoning questions. Similarly, robotic manipulation is more visually diverse and in the number of objects, but also suffers from a lack of diversity in the kinds of queries it needs to perform.

I get that the advantage of this method over pure NS approaches is that you do not need to manually define a domain-specific language, but the comparisons don't really any improvement over them. The claims about universality would be stronger if the domains were less constrained, showing for instance that this method can deal with unusual domain groundings in a way that would be hard to anticipate for a designer for NS methods.

Kind of a minor point, but I think the point on line 105 about VLMs not working on different domains such as 3D scenes and motion sequences is pedantic. Models not trained on 3D scenes and motion sequences of course don't generalize to these, but one could train a network on these domains as well. For instance: Gato (S Reed 2022) is trained on multiple kinds of modalities.

"Flamingo" in Table 6 should REALLY be stated to be Open-Flamingo. This is especially confusing since the paper uses the citation when it names the method. But it's not using the Flamingo model, it's using an open-source alternative to the model not Flamingo itself, which is very confusing/misleading.

A couple of unanswered questions I had (see below).

**Questions:**

In Table 6, is Neuro-FOL trained on the regular CLEVR task before being put in the held-out CLEVR tasks? What about [Open] Flamingo?

In general, could you more clearly state in the paper the training loop for Neuro-FOL. I sort of infer that it is trained end-to-end on the target tasks, but that the only parts of the network that are actually updated are the domain specific grounding modules? Is that correct?

How does your method respond to very rate concepts, if you need to finetune an MLP for every grounding function. Do you see cases where a concept has very few examples during training and that module performs poorly?

How often does the LLM fail to provide a valid program and how often does it fail to provide a correct program? Does an invalid program result in Neuro-FOL failing to run?

If the program is incorrect during training, does this hurt performance in eval?

How robust is the method in general to program failures?

In general, what are the failure cases of this method?

**Limitations:**

I don't think that the paper does much to address the limitations of their methodology (see above) which I think would add a lot of value to the paper.

---

> ### Author Rebuttal · Authors · 2023-08-08
>
> **Q: Sufficient data for training grounding modules.**
>
> A: Thank you for bringing up this point! We agree that Neuro-FOL may fail with insufficient data, which is a limitation for most learning systems trained solely on one given dataset. We provide three main perspectives here.
> 1. Neuro-FOL is notably more data efficient than end-to-end methods due to its modular structure (decomposing complex concepts into primitive ones).
> 2. To partially mitigate this issue, Neuro-FOL instructs LLMs to canonicalize concepts when reasonable (e.g., "in between" and "sandwiched" may be interpreted to "between"), which helps learning certain rare concepts.
> 3. It is possible and straightfoward to incorporate externally trained modules. For example, we can plug in Grounding DINO [1] for object detection. However, for some tasks, there are no off-the-shelf general visual recognition models (e.g., relations and human motions); therefore, we train grounding modules solely from the dataset. We fully agree that how to better integrate pretrained models by finetuning, or by incrementally learning new concepts, is important. And indeed as you mentioned, if Neuro-FOL sees a completely new concept during evaluation, it will fail, similar to other methods trained solely on the given data. We will clarify this limitation in the main text!
>
> [1] Liu, Shilong, et al. "Grounding DINO." 2023.
>
> **Q: Rare concepts.**
>
> A: As mentioned above, Neuro-FOL’s modularity enables data-efficient learning of rare concepts: given one datapoint each for “rainbow fire hydrant” and “rainbow sink”, end-to-end methods may have one datapoint each of the two concepts, while Neuro-FOL will have two datapoints for “rainbow”, and potentially leveraging data for “fire hydrant” and “sink” elsewhere. This is partially reflected in our data efficiency comparisons. Yet without enough data, Neuro-FOL’s performance will indeed suffer. For example, in ReferIt3D, the concept "clothes" appears only 18 times in training, and its classification accuracy is 0.207. Similarly: "cart", 84 times, accuracy 0.320. By contrast, "couch", 3640 times, 0.920, and "sink", 3028 times, accuracy 0.892.
>
> **Q: Diversity of evaluation domains.**
>
> A: We agree that the long-tail problem is important, and difficult. However our goal here is to integrate domain-specific concept learning with a unified, domain-independent reasoning framework, hence we defer the integration of few-shot learning methods to future work. We also note that some domains we evaluate on do have complex reasoning structures; in NR3D (Table 3) we see both visual and question diversity. Complex query examples include: “It's the middle towel out of the 3 that's all beside each other, disregard the only towel”, “If you are sitting in bed with your head against the headboard, it is the nightstand to your left.”
>
> **Q: Universality of FOL.**
>
> A: Thank you for raising this point. We do not claim to outperform prior neuro-symbolic methods as they represent "oracle" implementations for particular datasets. We instead focus on a unified framework across domains and tasks — with the benefits of neuro-symbolism, and minimizes the need for domain-specific knowledge.
>
> In the global response, we have also added code samples that highlight how Neuro-FOL’s unified reasoning modules enable flexible neuro-symbolic learning across domains.
>
> Additionally, in the CLEVR-Puzzle task (Section 4.2), we demonstrate Neuro-FOL in a setting where the language instruction cannot be parsed by the original CLEVR DSL, however, it can be represented by combining concepts with our general FOL, showing that Neuro-FOL can handle new tasks that are hard to anticipate for original DSL designers.
>
> **Q: Training data for held-out CLEVR tasks.**
>
> A: Yes, Neuro-FOL is trained on the original CLEVR with 70,000 images; Open Flamingo is trained on LAION 2B with 5 million image-text examples from Multimodal C4.
>
> **Q: Training loop.**
>
> A: Yes, the domain-specific functions (feature extractors and concept MLPs) are trained through backpropagation, as our executor is fully differentiable, and we use LLMs inference-only.
>
> **Q: Program generation.**
>
> A: LLMs occasionally fail to provide valid programs for complex natural language queries: (rate 0.102 for CLEVR-Humans, 0.088 for NR3D). An invalid program will cause Neuro-FOL to fail. In the global response, we further compare this with VisProg and ViperGPT (both use LLMs for program generation). Neuro-FOL produced syntactically valid programs for 100% of queries in CLEVR-RPM (c.f. ViperGPT's 40%). For semantic correctness, since we can not automatically label programs, we evaluate execution results.
>
> **Q: Effect of program correctness & program robustness.**
>
> A: If the program is incorrect, it is either (1) not executable, hence we lose a datapoint, or (2) semantically incorrect, which will harm performance. In our experience, Neuro-FOL programs have high semantic accuracy across tasks. For syntax errors, we can easily detect and resample. However, there is currently no way to recover from semantic errors. We believe a promising direction is to finetune the LLM from execution feedback.
>
> **Q: Failure cases.**
>
> A: In our experiments, FOL programs are typically correct, but the domain-specific functions are more difficult to learn. For example, in ReferIt3D, there are 607 categories, with many classes visually similar (e.g., cabinet and dresser). We see that Neuro-FOL classification accuracy for concept grounding is not very high. Hence, we can potentially rely on more classification labels, instead of only language instructions, to improve performance on this task. Additionally as mentioned, program failures and rare concepts also decrease performance.
>
> **Q: VLMs across domains & Open-Flamingo.**
>
> A: Thank you for the suggestions, we will clarify!
>
> **Q: Limitations.**
>
> A: We agree that an extended discussion section tackling limitations and future work will be helpful! We will add this to the main text.

---

> > ### Author Response · Authors · 2023-08-19
> > **Happy to answer any further questions**
> >
> > Dear Reviewer DugZ,
> >
> > Thank you for reviewing our submission. We have posted our response per your suggestions and questions. We are happy to discuss with you and answer any further questions. As the deadline for discussion is approaching, we very much look forward to your feedback.
> >
> > Thank you,
> >
> > Authors

---

> > ### Comment · Reviewer_DugZ · 2023-08-21
> > **Response**
> >
> > Sorry for the late reply.
> >
> > I think the response answered a lot of my questions. Given that and that I liked the paper to begin with, I will increase my score by one.

---

> > > ### Author Response · Authors · 2023-08-22
> > > **Thank you**
> > >
> > > Dear Reviewer DugZ, thank you again for your helpful comments and for reviewing our response!   -Authors

---

### Official Review · Reviewer_B8BR · 2023-07-05

**Soundness:** 2 fair
**Presentation:** 3 good
**Contribution:** 3 good
**Rating:** 5
**Confidence:** 4

**Summary:**

The paper presents an approach for solving visual reasoning tasks across multiple domains such as 2D image QA and robotic object manipulation. The approach (Neuro-FOL) uses an LLM to generate a first-order logic (FOL) program that is executed with a combination of learned and predefined modules. One aim is to take advantage of a pre-trained LLM (e.g., GPT 3.5) to generalize to new visual reasoning tasks at inference time. Neuro-FOL is evaluated on a diverse set of benchmarks including the CLEVR 2D VQA tasks, ReferIt3D 3D spatial reasoning tasks, HumanMotionQA tasks, and the CLIPort object packing tasks. Results demonstrate that Neuro-FOL is competitive with and sometimes outperforms strong baselines.

**Strengths:**

- The problem of grounding concepts across different domains (e.g., 2D images vs. 3D scenes) is an important problem because many concepts are naturally defined across these domains. This problem is well motivated in the introduction.
- The ability of Neuro-FOL to learn new concepts that are generated by the LLM during training is neat. Practically speaking, this should save designers the time required to define these concepts.
- The paper evaluates Neuro-FOL on a diverse set of domains including 2D image QA, 3D scene QA, motion capture QA, and robotic manipulation, which demonstrates the generality of the proposed approach. Furthermore, Neuro-FOL has strong performance in all tasks, suggesting that it will similarly do well in other domains.
- The paper is well written and easy to follow, but some details are missing as discussed below.

**Weaknesses:**

- The abstract and introduction start with the grand vision of grounding the concept “left” in multiple domains. However, from the experimental evaluations, it is unclear if such “domain-independent” reasoning is (a) learned and (b) useful.
   - (a) Are concepts learned in a domain-independent manner? In other words, is Neuro-FOL jointly trained on datasets from multiple domains? These details are not clearly stated in the current manuscript.
   - (b) The experimental results do not clearly demonstrate if “domain-independent” grounding is useful. A specific experiment to test this would be training Neuro-FOL independently on different domains and then comparing the performance of Neuro-FOL training jointly. Such an experiment does not appear in the paper. Thus, I do not see support of these claims.
- The proposed approach differs from LLM + API based methods (e.g., Gupta and Kembhavi, 2023) in two ways: (a) the use of first-order logic and (b) not requiring a predefined set of modules (i.e., concept networks in Neuro-FOL). It is unclear which of these two design concepts is most important. We can imagine a world in which (a) does not matter but (b) is critical. Alternatively, both (a) and (b) might be important. The experimental evaluations do not disentangle these two features. Thus, it is unclear where the important novelties in the method lies. Note: the ViperGPT experiments do not disentangle this question. Thus, it would be helpful if the authors could comment on this further.
- While learning concept MLPs through Neuo-FOL training has the benefit that concepts do not need to be predefined, this may also be a disadvantage because there may only be a small amount of data for a given concept. For example, a given dataset may have many questions asking about “left” and “right” and may only have a few asking about “in between” or “sandwiched between”. It is unclear how the proposed approach would generalize to such situations.
- L257: “Notably, Neuro-FOL is able perform on all domains, instead of requiring specific models for each setting.” This claim is misleading, as components of Neuro-FOL are trained for specific domains.

**Questions:**

1. How does Neuro-FOL performance compare when training jointly on multiple domains vs. independently on individual domains?
2. Are both (a) using FOL and (b) not having a predefined set of concepts important?
3. How does Neuro-FOL perform on concepts that are not well represented in the training data?

**Limitations:**

The limitations are briefly discussed, but additional details could be provided (perhaps based on answers to the questions above). The broader impacts are discussed in the appendix.

---

> ### Author Rebuttal · Authors · 2023-08-08
>
> **Q: Domain-independent reasoning & joint training.**
>
> A: We realize that there may be confusion in our descriptions about "domain-specific" vs. "domain-independent" and "reasoning" vs. "grounding." Here, we clarify that Neuro-FOL is a framework of "domain-independent reasoning" (composed of an LLM interpreter and FOL executor) and "domain-specific grounding." Concretely, we learn the grounding of the word "left" separately in different domains, because they are grounded on different modalities: 2D, 3D, temporal sequences, etc. We agree that learning a single representation for the concept "left" that is shared across different modalities is an interesting and important direction.
>
> To directly answer your questions, (a) concepts are learned through a “domain-specific” grounding module based on input modality, and (b) we do not conduct “domain-independent” grounding, instead we conduct domain-independent *reasoning* through a LLM interpreter and FOL executor. All experiments are trained on different domains, but Neuro-FOL enables the same domain-independent reasoning mechanism to work on all tasks.
>
> We do not conduct joint training across domains as (1) the domain-independent LLM interpreter and FOL executor are not trained, and (2) the domain-specific grounding modules can not be easily shared, as at the end they operate on different modalities. Because Neuro-FOL can be trained, a promising future direction is to finetune the LLM if we had access to the model weights and sufficient compute, and hence improve program generation and transfer reasoning across domains. We will clarify this in the main text!
>
> In the global response, we have also added code samples that highlight how Neuro-FOL’s shared domain-independent components enable flexible adoption of neuro-symbolic learning for new domains and tasks.
>
> **Q: Difference from LLM + API methods & importance of (a) usage of FOL and (b) no requirement of predefined concepts.**
>
> A: Thank you for suggesting the disentanglement between the "reasoning primitives" and "learnable concepts!" The most important contribution we want to highlight in this paper is indeed (b). In many domains (e.g., human motion and robotic manipulation), it is important to have systems that can learn new concepts from available data outside of predefined functions, as mentioned in your review. Our system allows for this, and learns new concepts instead of being constrained to an inference only system. Regarding (a), it is definitely possible to extend our language with more primitives (e.g., the ones used by ViperGPT and VisProg). In this work, we chose FOL primarily because it is general, expressive, and allows us to build an end-to-end differentiable computation graph while executing the programs across all domains.
>
> Please also see our added experiments on VisProg in addition to ViperGPT in the global response that analyzes differences between LLM + API baselines and Neuro-FOL.
>
> **Q: Sufficient data for concept learning & performance on not well represented concepts.**
>
> A: Thank you for bringing up this point! We agree that Neuro-FOL may perform poorly when there is insufficient data for training. For example, in the ReferIt3D task, the concept "clothes" is not well represented and appears only 18 times in the train set, and its classification accuracy in the test set is 0.207; concept "cart" occurs 84 times with accuracy 0.320. By contrast, more commonly seen concepts such as "couch" occur 3640 times with accuracy 0.920, and "sink" occurs 3028 times with accuracy 0.892. We note that this is not a disadvantage particular to our system, as all learning systems will face this problem when they are trained solely on the given dataset. We provide three main perspectives to this problem.
>
> First, Neuro-FOL is notably more data efficient than end-to-end methods due to its ability to decompose learning into modular networks and generate more training data for each individual concept. For example, given one training datapoint each for “rainbow fire hydrant” and “rainbow sink”, end-to-end methods may have one datapoint for each of the two rare concepts, while Neuro-FOL will have two datapoints for “rainbow”, and potentially more data from “fire hydrant” and “sink” learned elsewhere. This is partially reflected in the data efficiency comparisons in Table 4 and 5.
>
> Second, to partially mitigate this issue for Neuro-FOL, we instruct the LLM interpreter to canonicalize the concepts when reasonable, hence concepts such as “in between” and “sandwiched between” may both be merged to the “between” concept by the LLM.
>
> Third, it is possible and straightfoward to incorporate externally trained models in Neuro-FOL. We can integrate pretrained models as domain-specific grounding modules, for example, directly plugging in Grounding DINO [1] for object detection. For some tasks, there are no off-the-shelf general visual recognition models (e.g., relations, 3D concepts, human motion). Therefore, we train our concept groundings by learning solely from the dataset. We definitely agree that how to better integrate other pretrained models by finetuning, or by incrementally learning new concepts, is important. Thank you for the feedback! We will clarify this in an extended discussion section in the main text.
>
> [1] Liu, Shilong, et al. "Grounding DINO." 2023.
>
> **Q: Clarification of Neuro-FOL’s ability to perform across domains without requiring specific models.**
>
> A: Thank you for catching this, we will update this to without “requiring domain-specific *reasoning modules*”. Our intended goal was to highlight that Neuro-FOL’s domain-independent LLM interpreter and FOL executor are re-used on all domains, in contrast to prior works which require new models to be defined from scratch.
>
> **Q: Limitations.**
>
> A: We agree that an extended discussion section tackling limitations and future work will be helpful! We will add this in the main text.

---

> > ### Author Response · Authors · 2023-08-19
> > **Happy to answer any further questions**
> >
> > Dear Reviewer B8BR,
> >
> > Thank you for reviewing our submission. We have posted our response per your suggestions and questions. We are happy to discuss with you and answer any further questions. As the deadline for discussion is approaching, we very much look forward to your feedback.
> >
> > Thank you,
> >
> > Authors

---

> > > ### Comment · Reviewer_B8BR · 2023-08-21
> > > **Thank you for your response**
> > >
> > > Thank you for your response. My questions have largely been resolved. I will update my score accordingly.

---

> > > > ### Author Response · Authors · 2023-08-22
> > > > **Thank you**
> > > >
> > > > Dear Reviewer B8BR, thank you again for your helpful comments and for reviewing our response!   -Authors

---

### Official Review · Reviewer_xPiC · 2023-07-07

**Soundness:** 3 good
**Presentation:** 3 good
**Contribution:** 3 good
**Rating:** 7
**Confidence:** 5

**Summary:**

This paper proposes an approach for general-purpose language grounding across a variety of domains and tasks. The approach first uses an LLM to generate a domain-general program, which can be implemented across different domains using domain-specific functions, represented as neural networks (and domain-general implementations of certain FOL predicates). Experiments are conducted on a variety of domains, including image understanding, video understanding, and robotic manipulation.

**Strengths:**

The approach is compelling and relatively original -- it is especially original in combining a single unified formal representation across a variety of tasks. It is obviously addressing an important problem, and having modular approaches like these is important for investigating failure cases.

The approach is quite clearly detailed, except for details in training.

Experiments are very thorough. I appreciate evaluating on human data. I would have liked to see more analysis in model performance.

**Weaknesses:**

My main concern is that modularizing the approach into essentially two components: (a) language -> FOL, and (b) FOL + perception -> answer/action, is glossing over nuances of language use that may require interaction between the language and perception aspects. For example, the implementation of "left" may be dependent on the identities of the objects participating in the spatial relationship: there may be a canonical "left" side of an object, given its affordances (e.g., the "left" of a fridge may be the area to its left when one is standing facing the fridge, regardless of the actual perspective being taken in the 3D question answering task); it seems the current approach would not be able to handle this context-dependence of "left", unless the domain-specific implementation of "left" was somehow able to encode all of the information necessary to make this inference (that a particular region is a fridge, and that a fridge has a canonical "left"). Another place where this comes up is that the meaning of adjectives can be highly context dependent in natural language use: e.g., the meanings of color terms depend strongly on context (as a specific example: let's say there is a very pale blue object and a very pale red object -- when hearing "blue", one would most likely choose the pale blue object. But if the same very pale blue object is paired with a more vibrant blue object, "blue" is most likely going to refer to the more vibrant one). In general, I'd be interested to hear discussion about this particular problem and how (if) the proposed approach could implicitly capture these more direct relationships between language use and perception, or if FOL acts as a bottleneck which removes any ambiguity in sentence meaning (it seems this is the goal in its design -- even the program \iota x blue(x) ^ sphere(x) in L132 may be ambiguous if there are multiple blue objects of varying shades!).

I also would have liked to see more in-depth analysis of the approach. In particular, what's left for tasks where there is still a lot of room for improvement -- basically everything except CLEVR? Because the approach is modular, it seems there is a lot of room for analysis in how different parts of the approach might be failing -- are the programs incorrect? (What causes a program to be inexecutable?) Are the learned domain-specific functions erroneous?

Small nitpicks:
* Having a consistent example in Figure 2 would be nice.
* I'd suggest putting the FOL programs in Figure 3

**Questions:**

* Is using MLP implementations of domain-specific functions limiting in some ways? I.e., are there functions which, in experiments, are more difficult to train than others?
* Are the implementations of the FOL primitives hardcoded (e.g. as "min" for "and")?
* How are the domain-specific functions trained? There are few details on the training process, but I am assuming as it is end-to-end differentiable that they are trained using LLM-generated programs on some training data
* How is the arity determined for concepts? Is it just "proposed" by the LLM during training?
* I was confused about the presentation of Table 4 and 5 -- what is a data efficiency setting? Is this just the amount of training data used (presumably to train the domain-specific functions)?

**Limitations:**

There wasn't much discussion of limitations. I would have liked to see discussion wrt. analysis of the proposed approach, and the potential limitation of completely separating the language and perception aspects of these tasks by a single FOL program.

---

> ### Author Rebuttal · Authors · 2023-08-08
>
> **Q: Language and perception.**
>
> A: Thank you for bringing up these examples! There are indeed nuances that require interactions between language and vision. Similar to what you suggested, there are two candidate approaches: 1) interpreting language semantics in a context manner, by considering, e.g., the comparison classes for certain adjectives, and 2) relying on grounding modules for learning context-dependent grounding. In this paper, we choose 2) because our main focus is on integrating the unified reasoning framework and domain-specific concept learning. In general, we think option 2) has the advantage that the language interpretation part can be solely trained on text data, which is of a larger scale than vision-language data.
>
> Reference frames in spatial relations are handled in our work explicitly — they are specified in instructions in our 3D dataset. For example, given “Facing the cabinets, pick the object to its left”, Neuro-FOL generates `view(λx: cabinets(x)) and iota(λx: object(x) and left(x, iota(λy: cabinets(y))))`
> and the grounding of concepts will explicitly model the viewpoint of the agent. The view function is learned as a generic high-arity function, instead of specific ways to transform the point cloud. You are correct the semantics of “left side” is object-specific. Neuro-FOL will learn a “left_side” concept that captures a canonical left side of the object from the data distribution, since the relation grounding module takes object features as input too.
>
> Contexts (e.g., comparison classes) in adjective grounding is another important aspect. In the color example, Neuro-FOL can partially handle this by modeling the "degree" of whether each entity satisfies the concept. For example, to find *the* blue object,” the FOL execution assumes only one single object will be selected via "softmax", which partially handles the ambiguity. Concretely, suppose we have three objects, a pale blue, pale red, and vibrant blue object. To find *the* blue object, it would choose the “bluest” one—the vibrant blue object if compared to both others. This will also work to a certain extent, for compositional concepts such as `blue(x) ^ sphere(x)`. In other cases, since our object encodings do contain contextual information; therefore, most of the time we rely on learning from datasets to resolve such grounding.
>
> In sum, Neuro-FOL relies on context-dependent grounding modules to handle these cases. Though we do not build in pragmatic reasoning frameworks such as RSA [1], it is a direction we are excited about. We are very interested in tasks with ambiguous language and extending Neuro-FOL to be truly probabilistic, modeling speaker intents, and leveraging feedback from perception execution. Thank you again for the great feedback!
>
> [1] Goodman, Noah D., et al. Pragmatic language interpretation as probabilistic inference. Trends in cognitive sciences (2016).
>
> **Q: Analysis of Neuro-FOL.**
>
> A: Indeed, Neuro-FOL’s modularity enables error attribution. For more difficult tasks (3D, human motion, robotics), the FOL programs are typically correct, but the domain-specific functions are more difficult to learn. For example, in ReferIt3D, there are 607 categories, with many classes visually similar (e.g., cabinet and dresser). We see that Neuro-FOL classification accuracy for concept grounding is not very high. Hence, we can potentially rely on more classification labels, instead of only language instructions, to improve performance on this task.
>
> For syntax errors in LLM-generated programs, we can easily detect them and resample new programs. However, there is currently no way we can recover from semantic errors in the programs. We currently use GPT-3.5 in our experiments, but anticipate that GPT-4 would yield stronger performance; however, we were not able to leverage GPT-4 due to cost. Additionally, a promising future direction is to finetune the LLM from execution feedback.
>
> **Q: MLP implementations.**
>
> A: Thank you for bringing up this point! We do assume that MLPs can realize all domain-specific functions, as they take learned features as inputs, which are also jointly trained. This can occasionally be limiting if some functions are significantly more difficult to learn. In HumanMotionQA, for example, “action” concepts yield accuracy 0.637, while “body part” concepts only yield 0.495. Actions involving motion cues in multiple frames are easier to learn.
>
> We can definitely extend such MLP implementation to use more complex NNs, which likely require more training data. Additionally, one can initialize different networks for different concepts if they have prior knowledge on the task. We can also directly integrate pretrained visual models as domain-specific grounding modules.
>
> **Q: Function implementations.**
>
> A: Yes, Neuro-FOL requires a small but general set of functions for execution, listed in Table 1. These functions are either general logic and numeric operations (e.g., counting, exists), or functions that handle inputs and outputs (e.g., return a text description, execute an action).
>
> **Q: Training process.**
>
> A: Yes, the domain-specific functions are trained through backpropagation from the downstream loss functions, as our executor is fully differentiable.
>
> **Q: Arity for concepts.**
>
> A: Yes, both concept names and arities are proposed by the LLM. For example, given `above(x, y)`, we will treat `above` as a binary relation.
>
> **Q: Data efficiency setting.**
>
> A: Yes, the percentages in Tables 4 and 5 indicate what percentage of train data compared to the full dataset. For example, for Table 4, we train on 0.5% (329 examples), 1.5% (987 examples), etc., and show improvements over end-to-end methods.
>
> **Q. Figure edits.**
>
> A: Thank you for the suggestions! We will update the paper accordingly.
>
> **Q: Limitations.**
>
> A: We agree that an extended discussion section of limitations and future work will be helpful, and will add it to the main text incorporating all comments during review.

---

> > ### Author Response · Authors · 2023-08-19
> > **Happy to answer any further questions**
> >
> > Dear Review xPiC,
> >
> > Thank you for reviewing our submission. We have posted our response per your suggestions and questions. We are happy to discuss with you and answer any further questions. As the deadline for discussion is approaching, we very much look forward to your feedback.
> >
> > Thank you,
> >
> > Authors

---

> > ### Comment · Reviewer_xPiC · 2023-08-21
> >
> > Thank you for your rebuttal and apologies for not responding earlier. It has answered my questions and I would still like to see this paper accepted.

---

> > > ### Author Response · Authors · 2023-08-22
> > > **Thank you**
> > >
> > > Dear Reviewer xPiC, thank you again for your helpful comments and for reviewing our response!   -Authors

---

### Official Review · Reviewer_2ce2 · 2023-07-07

**Soundness:** 3 good
**Presentation:** 3 good
**Contribution:** 3 good
**Rating:** 6
**Confidence:** 5

**Summary:**

The manuscript proposes a framework that leverages an LLM to map queries to executable first-order-logic programs, with domain-specific grounding functions. The manuscript includes experiments on multiple tasks and domains.

**Strengths:**

The paper is mostly well-written.

The paper considers an interesting topic, under compelling methodology (i.e., neuro-symbolism).

The presentations of visualisations and results are mostly clear.

**Weaknesses:**

Section 1 (Introduction) / Figure 1 — One has to be particularly careful in applications that involve some (other) embodiment, as concepts like “left” could adopt an alternative reference frame, beside just the egocentric one. How does the approach incorporate this reasoning?

Section 1 (Introduction; L49-51) — Whereas no domain-specific language is required at the reasoning level, there still needs to be lexical alignment with the concepts supported by the domain-specific grounding modules. Furthermore, the approach does still need predefined built-in functions (Table 1) and  domain-specific concept names (L158).

Section 3.2 (L175-177) — Let’s be exceedingly explicit here. The manuscript states that the function implementations do not require manual definition in code. How are they generated? What are all the core FOL programs? What are their mappings from concepts initialized by the LLM?

Table 4 / Table 5 — The data percentages seem to be chosen arbitrarily. How were the percentages decided on? What happened for, e.g., 25% and 50%?

**Questions:**

N/A — see above

**Limitations:**

The manuscript does not include any sections on Limitations or Societal Impact.

---

> ### Author Rebuttal · Authors · 2023-08-08
>
> **Q: Alternative reference frame for “left”.**
>
> A: We thank you for bringing up this point! In domains where we need reference frames to disambiguate concepts like “left” (e.g., in 3D), the grounding of such concepts will explicitly consider the viewpoint of the agent — usually specified in the language query. For example, in the SR3D dataset, given the query “Facing the cabinets, pick the object to its left”, Neuro-FOL will generate the following FOL formula:
> ```view(Object, lambda x: cabinets(x)) and iota(Object, lambda x: object(x) and left(x, iota(Object, lambda y: cabinets(y))))```.
> The view function is learned as a generic high-arity function, without requiring any specific implementation to change the actual viewpoint of the pointcloud. In domains where no viewpoint is specified in language, such as the 2D domain, we assume an egocentric reference frame. We will clarify this in the main text.
>
> **Q: Clarifying concepts supported by grounding modules.**
>
> A:  We clarify that Neuro-FOL does not depend on a predefined set of lexical concepts and this is an important advantage of our system compared to earlier work on neuro-symbolic learning. To achieve this, we rely on LLMs to *automatically* extract concept names from natural language instructions in the dataset; Neuro-FOL then initializes domain-specific grounding modules (in our work, MLPs) accordingly to learn the grounding of these concepts from training data. In other words, the concepts and grounding modules are *not* predefined and can support execution of any given query. We note that *domain-specific concept names* means that these concepts are grounded to particular domains (e.g., 3D point clouds) and not shared across different datasets, but they are not manually defined.
>
> Neuro-FOL does require a minimal but general set of predefined FOL functions. Important to our goal of a general neuro-symbolic framework without any predefined domain-specific functions, Neuro-FOL only requires the FOL functions in Table 1 to reason across 2D, 3D, temporal, and robotic manipulation domains. These functions are either general logic and numeric operations (such as counting, forall, exists), or functions that handle inputs and outputs (e.g., return a text description of object categories, or execute an action output by other modules).
>
> For a new task, prior neuro-symbolic approaches require designing new domain-specific languages to support necessary operations and writing differentiable implementations for each function. In contrast, Neuro-FOL enables easy integration, while retaining all the benefits of neuro-symbolic learning. For a new task, we can simply define a neuro-symbolic system with the following code:
>
> ```
> domain = make_domain(ALL_LANGUAGE)
> parser = GeneralizedFOLPythonParser(domain)
> executor = GeneralizedFOLExecutor(domain, parser)
>
> scene_graph = SceneGraph{2D/3D/temporal}(THIS_INPUT_SCENE)
> with executor.with_grounding(scene_graph):
>     parsing = parser.parse_expression(THIS_INPUT_TEXT)
>     execution = executor.execute(parsing)
>     loss = loss(execution, gt)
> ```
>
> By passing in all language input, a scene (2D, 3D, temporal, etc), and a text query, Neuro-FOL will automatically define learnable domain-specific grounding modules, generate the FOL program, and execute domain-independent functions differentiably, such that you can simply apply any downstream loss as you would an end-to-end method. We believe this is an exciting improvement over prior works that will increase accessibility and flexibility of neuro-symbolic learning. We will clarify this in the main text!
>
> **Q: Clarifying function implementations.**
>
> A: Here, what we mean is, for all concepts extracted by LLMs, we use a generic mechanism to define the corresponding grounding module. For example, if a concept is unary (e.g., ```red(x)```), the grounding module will be a MLP layer that maps object representations to a scalar in [0, 1]. For concept ```beside```, the LLM specifies that the concept takes in features of two entities, and hence Neuro-FOL creates an MLP that operates on binary features.
>
> More specifically, given the LLM-interpreted program ```exists(Object, lambda x: dresser(x) and beside(x, iota(Object, lambda y: cabinet(y)))```, ```exist``` and ```iota``` are domain-independent FOL functions, which take as argument outputs from domain-specific functions of ```dresser```, ```cabinet```, and ```beside```.  The FOL functions have built-in implementations and are listed in full in Table 1, with implementations in Appendix A.1. The domain-specific concept grounding functions do not require manual definitions in code, and their implementations are generated using the generic mechanism described above. We will clarify in the main text.
>
> **Q: Data efficiency percentages.**
>
> A: For Table 4 and 5, we followed experiment settings of data percentages reported in prior state-of-the-art neuro-symbolic methods — NS-CL and NS3D. In Table 4, with 10% of train data, only small improvements upon end-to-end methods were seen, and hence additional percentages were not considered. For Table 5, we similarly followed prior work; due to your suggestion, we have additionally added experiment results for 25% and 50%. As expected, our work shows larger improvement upon MAC at more data efficient settings.
>
> |              | Pref-def.  |  10%  |  25%  |  50%  |  100% |
> | -----------  | ---------- | ----- | ----- | ----- | ----- |
> | NSCL         | Yes        | 0.989 | 0.992 | 0.994 | 0.992 |
> | TbD-Net      | Yes        | 0.541 | 0.560 | 0.912 | 0.991 |
> | MAC          | No         | 0.673 | 0.940 | 0.974 | 0.989 |
> | Neuro-FOL    | No         | 0.941 | 0.991 | 0.992 | 0.996 |
>
> **Q: Limitations and societal impact.**
>
> A: We include discussion in Section 4.2, and broader impacts in Appendix A.3. We agree that an extended discussion section tackling both limitations and broader impact will be helpful! We will add an additional section in the main text taking into account all comments.

---

> > ### Comment · Reviewer_2ce2 · 2023-08-14
> > **Official comment by Reviewer 2ce2**
> >
> > I appreciate the authors' detailed responses to all reviews. I have no further comments, and I am satisfied with the answers to my questions; I will increase my score by one point.

---

> > > ### Author Response · Authors · 2023-08-19
> > > **Thank you**
> > >
> > > Thank you again for your helpful comments!  We are glad to hear that the concerns have been addressed.

---

### Author Rebuttal · Authors · 2023-08-08

We thank all the reviewers for their constructive feedback! We have added additional data efficiency experiments on CLEVR, baseline results from VisProg & ViperGPT for comparison to Neuro-FOL, statistics and analyses of Neuro-FOL performance from the HumanMotionQA and ReferIt3D tasks, and code examples to exemplify flexibility of Neuro-FOL usage.

Many reviewers suggested extending our discussion section in the paper, with which we fully agree. We will add a lengthened discussion section in the main text, with focus on limitations, future work, and broader impact, integrating all comments from your reviews.

Additionally, we’d like to highlight Neuro-FOL’s contribution over prior neuro-symbolic approaches. The existing methods require predefined “oracle” domain-specific functions for a given domain and task. For a new task, this means designing new domain-specific languages to support necessary operations and writing differentiable implementations for each function. This requires both domain-specific knowledge as well as neuro-symbolic background. In contrast, Neuro-FOL enables easy integration without either of the above requirements, while retaining all the benefits of neuro-symbolic learning. For a new task, we can simply define a neuro-symbolic system with the following code:

```
domain = make_domain(ALL_LANGUAGE)
parser = GeneralizedFOLPythonParser(domain)
executor = GeneralizedFOLExecutor(domain, parser)

scene_graph = SceneGraph{2D/3D/temporal}(THIS_INPUT_SCENE)
with executor.with_grounding(scene_graph):
    parsing = parser.parse_expression(THIS_INPUT_TEXT)
    execution = executor.execute(parsing)
    loss = loss(execution, gt)
```

By passing in all language input, a scene (2D, 3D, temporal, etc), and a text query, Neuro-FOL will automatically define learnable domain-specific grounding modules, generate the FOL program, and execute domain-independent functions differentiably, such that you can simply apply any downstream loss as you would an end-to-end method. We believe this is an exciting improvement over prior works that will increase accessibility and flexibility of neuro-symbolic learning.

Based on reviewer suggestions, we also ran experiments on VisProg in addition to the ViperGPT, in order to better analyze differences between LLM + API baselines and Neuro-FOL. Our method significantly outperforms baselines that leverage programs from LLMs, as Neuro-FOL has less predefined functions and more learned programs.

|                       | CLEVR-Ref  | CLEVR-Puzzles  |   CLEVR-RPM   |
| -----------       | ------ | ------ | -----|
| Neuro-FOL    | **0.94**   | **0.75**   | **0.87** |
| Flamingo 4-shot    | N/A   | 0.54   | 0.52 |
| Flamingo 8-shot    | N/A   | 0.57   | 0.52 |
| VisProg    | N/A   | 0.27   | 0.51 |
| ViperGPT    | N/A   | 0.34   | 0.04 |

We highlight some examples where Neuro-FOL improves upon these methods. For example, ViperGPT does not contain predefined programs for “left” and “right”, hence, when running inference, the LLM produces Python code to compare the pixel coordinates in order to find such relations, which does not take into account depth of the scene, sizes of the objects, etc. By contrast, Neuro-FOL is able to learn programs for “left” and “right” respectively for more faithful execution. As an additional example, VisProg often passes in full phrases to its predefined methods, such as ```LOC(image=IMAGE, object='small yellow metal cylinder')```, instead of composing modular concepts for "small", "yellow", "metal", and "cylinder" as Neuro-FOL does, which allows for better error attribution. In addition, for more complex domains such as 3D and temporal human motion sequences, it is unclear what predefined models should be used for LLM + API methods.

We thank you for all the helpful comments, and am happy to answer any follow-up questions!

---

### Decision · Program_Chairs · 2023-09-21

**Decision:**

Accept (poster)

**Comment:**

The paper is concerns with the use of LLMs for answering questions about data across different domains. It leverages LLMs to convert the questions to a logic-based domain-independent representation that is subsequently grounded in a domain of interest.

The paper has received 1 x Accept, 2 x Weak Accept, and 1 x Borderline Accept.

All reviewers appreciate the approach. The all hail the idea of using a domain independent formalism (“compelling methodology” (2ce2), “compelling and relatively original” (xPiC), “really clever and well motivated” (DugZ)). In particular, the reviewers find of value the ability to have general and formal representation of queries and questions, powered by LLMs. And then the ability to apply it to any domain afterwards with domain specific grounding. Further, the reviewers find the authors address an important and timely problem. The evaluation is compelling and convincing. Hence, the AC in consultation with the SAC accepts the paper to Neurips.

We will encourage the authors to incorporate some of the feedback given in the rebuttals. In particular, clarification re domain specific and domain independent, challenges and benefits in lack of enough domain data, etc.